# Leukaemic Presentation of Small-Cell Alk-Positive Anaplastic Large Cell Lymphoma in a Young Woman—Report of a Case with 9-Year Survival

**DOI:** 10.3390/medicina59091628

**Published:** 2023-09-08

**Authors:** Carlos Santonja, Daniel Morillo-Giles, Elena Prieto-Pareja, Carlos Soto-de Ozaeta, Cristina Serrano-del Castillo, Rocío Salgado-Sánchez, Ana-Wu-Yang Yi-Shi, Rebeca Manso, Socorro María Rodríguez-Pinilla

**Affiliations:** 1Departments of Pathology, Hospital Universitario Fundación Jiménez Díaz, 28040 Madrid, Spain; csantonja@fjd.es (C.S.); yi-shi_ana@hotmail.com (A.-W.-Y.Y.-S.); smrodriguez@fjd.es (S.M.R.-P.); 2Departments of Haematology, Hospital Universitario Fundación Jiménez Díaz, 28040 Madrid, Spain; daniel.morillo@quironsalud.es (D.M.-G.); eprietop@fjd.es (E.P.-P.); csoto@fjd.es (C.S.-d.O.); cserranoc@fjd.es (C.S.-d.C.); rocio.salgado@quironsalud.es (R.S.-S.)

**Keywords:** anaplastic large cell lymphoma, leukaemic, ALK positive survival, case report

## Abstract

Anaplastic large cell lymphoma (ALCL) with leukaemic presentation (either ab initio or along the course of the disease) has been rarely reported. Irrespective of ALK expression in the neoplastic cells, it features a dismal prognosis. We report a rare case of leukaemic, small cell variant ALK-positive ALCL with 9-year survival in a young woman who was treated upfront with corticosteroids and standard chemotherapy, and review thoroughly the previously published cases. Such an unexpected, good outcome hints at the existence of different clinical subgroups in the leukaemic variant of ALK-positive ALCL.

## 1. Case Report

A 24-year-old woman from Costa Rica presented in February of 2014 with malaise, fever, and disseminated lymphadenopathy. No cutaneous lesions or hepato-splenomegaly were detected. The leukocyte count was 86 × 10^9^/L with 75% abnormal lymphoid cells. Erythrocytes, platelets, and routine serum chemistry values were within normal range, except for a mildly increased LDH (552 IU/L, normal range 230–460). Giemsa-stained peripheral blood (PB) smears disclosed small to medium-size lymphoid cells with high nuclear-cytoplasmic ratio and distinctly convoluted, floret-like nuclei (Figure 1). On flow cytometry of PB, CD3, CD4 and CD25 were detected in the neoplastic cells, which were negative for CD8, CD5, CD7, TdT, CD34 and CD30. In sections from a cell block prepared from PB buffy coat there was no expression of Foxp3, TCL-1 by immunohistochemistry (IHC) or Epstein-Barr Virus encoded small RNA (EBER) by in situ hybridization. Based on morphologic, immunophenotypic and clinical grounds, a number of diagnostic options were regarded as highly unlikely, namely T-cell large granular lymphocytic leukaemia, aggressive NK-cell leukaemia, EBV-related T/NK-cell lymphoproliferative diseases, T-cell prolymphocytic leukaemia and Sezary syndrome (1). Negative results of serological studies for Epstein-Barr virus, cytomegalovirus and human T-lymphotropic virus 1/2 were obtained, the latter excluding adult T-cell leukaemia/lymphoma. A tentative diagnosis of PB involvement by T-cell lymphoma (not otherwise specified) was made, and the patient received systemic corticosteroids as pre-phase treatment. An inguinal lymph node excisional biopsy (Figure 1) showed a proliferation of small neoplastic cells, in a focally sclerotic background, with rare eosinophils. A distinct second population of larger cells was present in irregular nodules and scattered in subcapsular, intrasinusoidal, and perivascular location. IHC demonstrated CD3 in the small neoplastic cells with weak expression of CD30, CD25 and ALK (nuclear and cytoplasmic). Conversely, the large neoplastic cells lacked CD3 and were strongly CD30, ALK (nuclear and cytoplasmic) and CD25-positive. FISH studies on PB smears revealed a t(2;5) translocation. Cytogenetic studies could not be performed. Atypical lymphoid cells with convoluted nuclei were seen in bone marrow (BM) aspirate cytology, and a BM core biopsy showed interstitial infiltration by small lymphocytes (positive for CD3, CD30 and ALK), and rare large cells with horseshoe-shaped nuclei. A diagnosis of leukaemic, small cell variant of ALK-positive anaplastic large cell lymphoma (ALCL) was made. After the first cycle of chemotherapy with cyclophosphamide, doxorubicin, vincristine, prednisone, and etoposide (CHOEP) there was a marked decrease in the number of PB leukocytes and lymphocytes; complete remission was achieved after 6 cycles. The patient is alive and free of disease 114 months after diagnosis.

Anaplastic large cell lymphomas are divided according to site of involvement and presence or absence of ALK translocation, into ALK-positive and ALK-negative systemic, primary cutaneous and breast implant-related ALCLs [1,2]. ALK-positive ALCL usually presents as a widespread disseminated disease involving lymph nodes and extranodal sites. Involvement of bone marrow (BM) and peripheral blood (PB] occurs in less than 10% of the cases. Morphologically, ALK-positive ALCL exhibits a broad spectrum, including classic (70%), lymphohistiocytic (5–10%), small-cell (5–10%), and sarcomatoid variants (1%). Hallmark cells, which are typically large with eccentric horseshoe- or kidney-shaped nuclei, are seen in all variants and are helpful to establish the diagnosis. A characteristic finding in the small-cell variant is the reciprocal intensity of the immunohistochemical expression of CD3, CD30, ALK and CD25 in the large versus the small neoplastic cells, a feature also seen in our case. This has been recently investigated by molecular-based approaches, with STAT5a allegedly leading to down-regulation of NPM-ALK mRNA and CD30 protein expression [3]. Although ALK-positive ALCL cases have a relatively good outcome, the small-cell leukemic variant is -irrespective of ALK-translocation status- an aggressive disease. A total of 35 cases of leukaemic, small cell variant of ALK-positive ALCLs have been previously reported [3,4,5,6,7,8,9,10,11,12,13,14,15,16,17,18,19,20,21,22] (Table 1) with limited follow-up time (from 1 to 63 months) in the surviving patients. All evidence points to an aggressive disease with a median overall survival of 12 months and a high tendency for central nervous system recurrences. For most patients who are fit enough to receive combination chemotherapy, ALCL is treated similarly to other nodal PTCLs, and CHOP has been the backbone for frontline treatment. Attempts to improve first-line treatment included the addition of other agents to CHOP. The German high-grade non-Hodgkin lymphoma study compared CHOP therapy with CHOEP therapy [23]. Among the 320 patients with PTCL enrolled, younger patients (<60 years) with normal LDH values had a significant improvement in outcome if they received CHOP plus etoposide compared with CHOP alone, with 3-year event-free survival of 75.4% versus 51%, although no difference in overall survival (OS) was observed. Accordingly, CHOP plus etoposide (CHOEP) is a very good therapeutic option in young, fit patients, like the long-term survivor hereby presented.

The need for allogeneic stem cell transplantation to improve overall survival has been recently stressed. Our case is clinically unique in that peripheral blood neoplastic lymphocytes declined dramatically on initial corticosteroid treatment, and all signs of disease vanished after the first cycle of chemotherapy. Moreover, the patient is alive and free of disease after 9 years, which makes her a distinct outlier if the overall survival rate of T-cell lymphomas presenting with PB involvement is considered. Although the ultimate physiopathologic reason for this is a matter for speculation, it may well hint at the existence of clinical subgroups among the leukaemic small cell variant of ALK-ALCL, as has been suggested for other variants of either ALK-positive and negative ALCL cases.

## Figures and Tables

**Figure 1 medicina-59-01628-f001:**
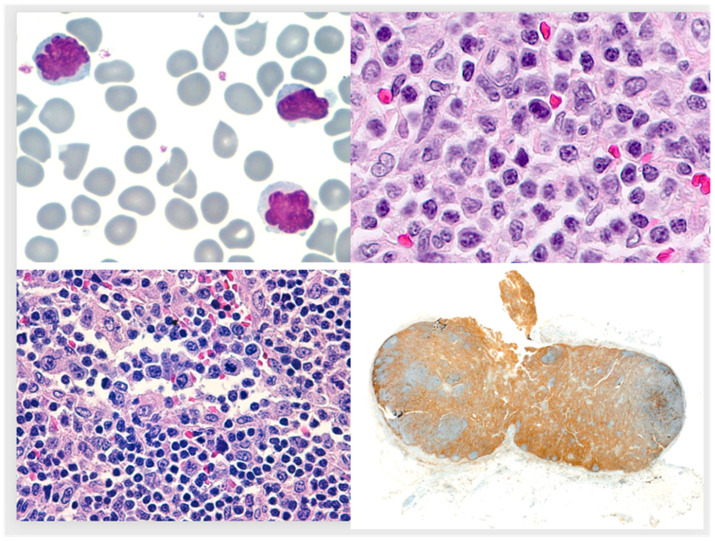
24 year old woman presenting with malaise, lymphadenopathy and marked lymphocytosis. Note floret-like appearance of the nuclei (**upper left** panel). Small to medium-sized lymphoid cells comprising most of the neoplastic tissue in an excised lymph node (**upper right** panel). Large, atypical cells in sinuses (**lower left** panel), which were also present in nodular areas in the lymph node (not shown). Low-power view of neoplastic lymph node with extensive ALK-positivity, stronger in the nodular area on the right-hand side (**lower right** panel); CD30 and CD25 (not shown) displayed similar findings.

**Table 1 medicina-59-01628-t001:** Clinical data and survival of previously reported cases of leukaemic small-cell variant of ALK-positive anaplastic large cell lymphoma.

Year, Author [Reference]	Sex/Age	Extent at Diagnosis (Other than PB or BM)	Treatment	Folllow-Up
1993, Kinney [18]	M/17	LNs, liver, skin	MEGA, autologous BM transplantation	Dead of infection at 1 mo
1999, Villamor [6]	M/36	LNs, liver, spleen	megaCHOP/ESHAP	Relapse at 2 mo
1999, Bayle [4]	F/10	Mediastinum	COPAD-M BCNU, vinblastine, cytarabine	1st relapse at 8 w/ A&W at 11 mo
1999, Bayle [4]	F/18	Mediastinum, cervical LN, skin	CHOP/CT followed by HSCT	1st relapse at 2 mo/ CR at 18 mo
1999, Bayle [4]	F/20mo	Axillary LN, liver, spleen	COPAD-M/ vinblastine-HSCT	1st relapse at 1 mo/ PDOD few mo later
1999, Bayle [4]	M/7	Skin, LNs, mediastinum	vinblastine, adriamycin, dexamethasone, methotrexate	SON
2000, Lesesve [7]	F/28	LNs, liver, spleen, skin	CHOP	DOD at 3 mo
2002, Awaya [19]	M/63	LNs, liver	CHOP	Dead of infection at 1 mo
2003, Onciu [20]	F/6	Lung, kidneys	doxorubicin, vincristine, prednisone; methotrexate, dexamethasone; lomustine, vinblastine, cytarabine; BM transplant	ANED at 17 mo
2003, Onciu [20]	F/9mo	LNs, spleen, liver, lung, skin	corticosteroids, cyclophosphamide, vinblastine; dexamethasone, daunorubicin, asparaginase, methotrexate; cytarabine, etoposide.	DOD at 9 mo
2003, Onciu [20]	M/10	maxillary sinus, LNs, CNS	methotrexate, ifosfamide, etoposide, dexamethasone; doxorubicin, vincristine, prednisone;methotrexate, cladribine, vinblastine.	DOD at 2 ys
2004, Kong [8]	F/32	LNs, mediastinum, liver, spleen, CNS	CHOP	DOD at 2 mo
2007, Grewal [5]	M/29	LNs, mediastinum, liver, spleen	daunorubicin, vincristine, prednisone; Hyper-CVAD; ICE; HSCT	DOD 1 mo after HSCT
2007, Grewal [5]	M/11	LNs, liver, spleen, colon/CNS	CCG-5941/D-ICE/ /cranial irradiation	DOD 3 mo
2007, Grewal [5]	F/59	Skin, liver	CHOP	DOD 1 mo
2008, Takahashi [22]	M/10	LNs, liver, spleen	dexamethasone, cytarabine, vindesine	Dead of infection before treatment end
2008, Sano [21]	F/23	LNs, liver, spleen, skin	CHOP, autologous BM transplant	Alive at time of report
2009 Nguyen [17]	M/26	LNs	cyclophosphamide, vincristine, doxorubicin, dexamethasone, intrathecal methotrexate and cytarabine	DOD 2.5 mo
2014, Spiegel [9]	F/10	LNs, skin, liver, spleen, lung	ALCL 99/CT + HSCT	Alive 23 mo after HSCT
2014, Spiegel [9]	M/13mo	LNs, skin, liver, spleen, lung, CSF	ALCL 99	Dead PD
2014, Spiegel [9]	F/20mo	LNs, liver, spleen	SFOP-HM 91/vinblastine +HSCT	Dead (toxicity)
2014, Spiegel [9]	M/11	LNs, skin, iver, spleen, lung	ALCL 99/HSCT	Alive 63mo
2014, Spiegel [9]	F/17	LNs, liver, spleen, lung	ALCL 99/LMB 96 protocol/vinblastine + HSCT	Dead PD
2014, Spiegel [9]	M/3	LNs, liver, spleen, lung, CSF	LMB 96 protocol/methotrexate, araC, etoposide/autoSCT/ICI + HSCT	Dead (toxicity)
2014, Spiegel [9]	F/6	LNs, skin, liver, spleen, lung	ALCL 99 protocol/vinblastine + CT + HSCT	Dead (toxicity)
2014, Spiegel [9]	F/12	LNs, liver, spleen	ALCL 99 protocol + vinblastine maintenance	Alive 34 mo
2014, Spiegel [9]	F/4	LNs, skin, liver, spleen, lung, kidney	ALCL 99 protocol/Crizotinib	Alive 15 mo
2014 Liu [10]	M/24	LNs, retroperitoneal	NS	NS
2017 Al-Ahmad [11]	F/57	LNs, liver, spleen	CHOEP	CR “Dec. 2016”
2017 Zecchini [12]	F/37	LNs	MACOP-B	NS
2018 Jiang [13]	M/47	LN	Brentuximab vedotin + CT	Clinical improvement
2019 Graetz [14]	M/16mo	LNs	ALCL99protocol/crizotinib	In remission
2020 Kundoo [15]	M/25	NS	None	DOD
2022 Noguchi [3]	M/10	liver, spleen	ALCL99protocol/alectinib/HSCT	NS
2022 Dutta [16]	M/68	None	NS	DOD
Present case	F/24	LNs	CHOEP	ANED 114 mo

Abbreviations and description of chemotherapy regimens: A&W, alive and well; ABMT, allogeneic bone marrow transplant; ALCL 99, anaplastic large cell lymphoma protocol 99: dexamethasone, cyclophosphamide, high-dose methotrexate, doxorubicin, vinblastine, ifosfamide, cytarabine and etoposide; ALK, anaplastic lymphoma kinase; ANED, alive with no evidence of disease; BCNU, Bis-chloroethylnitrosourea; BM, bone marrow; CCG-5941 Childrens Cancer Group: vincristine, L-asparaginase, prednisone and intrathecal methotrexate induction followed by consolidation with vincristine, etoposide (VP-16), cytosine arabinoside (Ara-C/VM-26), 6-thioguanine, high dose methotrexate and intrathecal methotrexate; CHOEP: cyclophosphamide, prednisone, adriamycin, vincristine, etoposide); CHOP, cyclophosphamide, adriamycine, vincristine, prednisone; CNS, central nervous system; COPAD-M: cyclophosphamide, vincristine, adriamycin, methotrexate; CR, complete remission; CSF, cerebrospinal fluid; CT, chemotherapy; D-ICE: dexamethasone, ifosfamide, cisplatin, and etoposide; DOD, dead of disease; ESHAP: etoposide, cisplatin, prednisone, cytarabine; F, female; HSCT, Haematopoietic stem cell transplantation; Hyper-CVAD, hyper-fractioned cyclophosphamide, vincristine, doxorubicin, and dexamethasone; ICE, ifosfamide, cisplatin, and etoposide; ICI: idarubicin-carboplatin-ifosfamide; LMB 96 Lymphoma Malignancy B 96: cyclophosphamide, vincristine, prednisolone and doxorubicin; LN, lymph node; LNs, multiple lymph nodes; M, male; MACOP-B, methotrexate, doxorubicin, cyclophosphamide, vincristine, prednisone, and bleomycin; mo, months; MEGA, dose intensive protocol at Vanderbilt University, i.e., cyclophosphamide, etoposide, doxorubicin, vincristine, bleomycin, methotrexate, leucovorin, prednisone; NS, not stated; PB, peripheral blood; PD, progressive disease; PDOD presumably dead of disease; SFOP-HM 91 French Society of Pediatric Oncology: cyclophosphamide, doxorubicin, etoposide, methotrexate, vincristine; SON, still on treatment; w, weeks; ys, years.

## Data Availability

The data presented in this study are available on request from the corresponding author.

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
