# Peer review of "Leukaemic Presentation of Small-Cell Alk-Positive Anaplastic Large Cell Lymphoma in a Young Woman—Report of a Case with 9-Year Survival"

_medicina, 2023, doi:10.3390/medicina59091628_

Round 1

Reviewer 1 Report

The authors describe a case of an ALCL, ALK positive, small cell variant with >9 year survival. The case is well-presented and interesting. I have only few minor comments:

1) Page 2, line 49/50: it should be given if ALK expression was cytoplasmic or nuclear or both (I suppose the latter).

2) Same page, line 68: The 5th edition of the WHO classification (Alaggio et al. Leukemia 2022) should be cited as well.

3) Same page, lines 89-91: This statement should be toned down a bit: "...may hint at..."

Author Response

The authors describe a case of an ALCL, ALK positive, small cell variant with >9 year survival. The case is well-presented and interesting. I have only few minor comments:

1) Page 2, line 49/50: it should be given if ALK expression was cytoplasmic or nuclear or both (I suppose the latter).

This has been done.

2) Same page, line 68: The 5th edition of the WHO classification (Alaggio et al. Leukemia 2022) should be cited as well.

This has been done.

3) Same page, lines 89-91: This statement should be toned down a bit: "...may hint at..."

This has been done.

Reviewer 2 Report

The authors  presented a case of ALCL in leukemic phase. The patient underwent CHOEP chemotherapy 6 cycles. Remission was achieved and he was reported to be a long-term survivor of this entity. 

My comments are the followings:

1. This case, although uncommon, is not exceedingly rare. While poor prognosis is expected, long term survival is not unlikely. There are survivors (although short follow up) in authors' literature review. 

2. The literature review is not complete. In 2009, there is a case report including review of 20 well-documented cases (Nguyen et al.). Some cases in the cases in that report by Nguyen et al. were not collected in the present manuscript.  In addition, there are some additional  cases in remission or survival in that report. 

3. The author should address an issue why this patient is doing better than others. Is CHOEP a better treatment regimen than others? 

The language quality is fine. Some very tiny revision may be needed if accepted for publication.

Author Response

The authors  presented a case of ALCL in leukemic phase. The patient underwent CHOEP chemotherapy 6 cycles. Remission was achieved and he was reported to be a long-term survivor of this entity. 

My comments are the followings:

  1. This case, although uncommon, is not exceedingly rare. While poor prognosis is expected, long term survival is not unlikely. There are survivors (although short follow up) in authors' literature review. 

We agree than some patients have survived longer than expected, but a 6-year survival is -to the best of our knowledge- not on record.

  1. The literature review is not complete. In 2009, there is a case report including review of 20 well-documented cases (Nguyen et al.). Some cases in the cases in that report by Nguyen et al. were not collected in the present manuscript.  In addition, there are some additional cases in remission or survival in that report. 

Many thanks for pointing out a missed reference; this has been included in the table and cited accordingly, and the pertinent references in Nguyen's table have also been included.

Review of Nguyen 2009 :

Case 1 in Nguyen's table (a 17 y/o male) corresponds to patient 3 in Kinney's 1993 table (https://pubmed.ncbi.nlm.nih.gov/8394652/) and has also been included in the table.

Case 2 in Nguyen's table (a 36 y/o male) was reported in 1996 by Anderson MM et al and has been omitted, since we are focusing on the small cell variant of anaplastic large cell lymphoma.

Cases 3-7 in Nguyen's table had already been included in our review.

Case 8 in Nguyen's table was reported by Meech et al in 2001 and does not seem to conform either morphologically or phenotypically to a standard small-cell variant of anaplastic large cell lymphoma and was therefore omitted.

Case 9 in Nguyen's table was reported by Awayael al in 2002 and has been included https://pubmed.ncbi.nlm.nih.gov/11891807/

Cases 10-12 in Nguyen's table were reported by Onciu et al 2003 and have been included

https://pubmed.ncbi.nlm.nih.gov/14560573/

Cases 13-16 in Nguyen's table had already been included in our review

Case 17 in Nguyen's table was reported by Sano et al 2008 and has been included

https://pubmed.ncbi.nlm.nih.gov/18705769/

Case 18 in Nguyen's table was reported by Takahashi et al 2008 and has been included

https://pubmed.ncbi.nlm.nih.gov/18776764/

Case 19 in Nguyen's table was reported back in 1988 by Fischer et al and appears to be a conventional anaplastic large cell lymphoma with leukaemicinvolvent , not a small-cell variant; it has therefore not been included.

Case 20 in Nguyen's table was reported by Dalal et al 2005 and corresponds to a conventional anaplastic large cell lymphoma with peripheral blood involvement, and has therefore not been included

  1. The author should address an issue why this patient is doing better than others. Is CHOEP a better treatment regimen than others? 

This has been done.

Round 2

Reviewer 2 Report

The authors added the references provided by the reviewer but decided to only focus on small cell variants. In fact, the included cases in review are not all small cell variants. For example, in the study by Spiegel et al, a small cell component was present in 6/9 cases.

Long term survival is possible. I think the key point is the scientific basis of such a clinical phenomenon, which is not adequately addressed. 

 Quality of English is acceptable.

Author Response

Dear Reviewer 2,

Thanks for reviewing once more our paper.

We did intend from the beginning to focus on the small cell variant of anaplastic large cell lymphoma, leukaemic type, although I am afraid we did not make it sufficiently clear either in the abstract or in the paper introduction; we have modified this accordingly. In our review we left therefore intentionally out those leukaemic REGULAR large cell anaplastic lymphomas. Regarding the paper by Spiegel, although perusing the table one gets the impression that 3 of the cases did not correspond to small cell variants, it is stated in paragraph one, under "Biological Findings", page 546 that "All patients had a high leucocyte count at diagnosis with a median level of 75 9 109/l [range, 30–120 9 109/l]. The proportion of circulating tumour cells among lymphoid cells ranged from 12 to 90%. The morphological features of these blood cells were diverse and represented by small cells with a notched, lobulated and dense nucleus or rare large basophilic vacuolated cells."  We regard the appearance of the abundant small cells in peripheral blood in our case (mimicking the floret cells of HTLV-1-related adult T-cell leukaemia/lymphoma) as a diagnostic pitfall, and have tried to convey this information to our potential readers.

As for the ultimate cause of our patient's inordinate long survival, we are unable to provide a scientific rationale, and must leave the question open. The final sentence of the paper has been worded to that end.